# The Effects of the Supercritical Extracts of *Momordica charantia* Linn., *Pistacia lentiscus*, and *Commiphora myrrha* on Oral Inflammation and Oral Cancer

**Mi Jeong Choi [1,*], Hwa Seung Yoo [2,*] and So Jung Park [3]**

[1]  Biomedical Biotechnology Research Institute Co., Ltd., Goyang 10326, Korea
[2]  East-West Cancer Center, Seoul Korean Medicine Hospital of Daejeon University, 32, Beobwon-ro 11-gil, Songpa-gu, Seoul 05836, Korea
[3]  East-West Cancer Center, Daejeon Korean Medicine Hospital of Daejeon University, 75, Daedeok-daero 176 beon-gil, Seo-gu, Daejeon 35235, Korea; vivies@hanmail.net
*   Correspondence: wbio2008@daum.net (M.J.C.); altyhs@dju.ac.kr (H.S.Y.)

**Abstract:** In this study, mixed extract samples (MPC-1–4) of natural plants, *Momordica charantia Linn.*, *Pistacia lentiscus*, and *Commiphora myrrha*, were prepared according to their respective extraction methods, and the efficacy of these samples for treating oral inflammation and oral cancer was investigated. As a result of the cell proliferation inhibition experiment, all samples (MPC-1 to 4) decreased the proliferation of oral cancer cell MC3 and HN22 cells in a concentration-dependent manner ($p < 0.01$). The survival rates of MPC-4 and MPC-1 were about 50% and about 80%, respectively, showing a difference according to the extraction method. In flow cytometry results, early apoptosis and late apoptosis of MPC-4 were 26.9% and 18.1%, respectively, indicating that apoptosis induction was the most effective. Although the induction effect was shown in other samples, the result was lower than that of MPC-4. As a result of confirming the regulation of the signaling pathway, it was confirmed that the expression of cleaved caspase 3 and Bak regulatory genes increased in a concentration-dependent manner in MC3 and HN22 cells ($p < 0.01$), thus inducing apoptosis in oral cancer cells. In addition, as a result of safety and Xenograft model experiments, it was found that MPC-4 had no toxicity to oral administration. These results suggest that the supercritical extract of *Momordica charantia Linn.*, *Pistacia lentiscus*, and *Commiphora myrrha* can be applied as a preventive and therapeutic agent for oral mucosa inflammation and oral cancer.

**Keywords:** oral inflammation; oral cancer; *Momordica charantia Linn.*; *Pistacia lentiscus*; *Commiphora myrrha*

## 1. Introduction

Cancer is the most threatening disease, occupying the number one cause of death, in South Korea. According to the e-Nara Index, the cancer incidence rate has been increasing every year, leading to a cancer death rate of 158.2 per 100,000 in 2019, an increase of 17.7 compared to 10 years ago, and the resultant side effects and toxicity problems due to cancer treatment still remain as problems that must be solved [1]. This means that oral health is very important to sustain a healthy human life. In particular, the oral cavity is a very important part of the body for humans in that it is a pathway for humans to consume nutrients necessary to maintain a sustainable life. Approximately 35–87% of chemotherapy patients experience stomatitis and resultant discomfort [2]. Stomatitis appears due to the cytotoxic agents used during chemotherapy, and causes ulcers, inflammation, and bleeding to the lips, gums, tongue, and mucosal tissues, leading to acute pain and food intake disorders [3]. In cancer patients receiving chemotherapy, stomatitis is one of the most commonly appearing complications, which reduces the patient's immune function and ability to resist against bacteria infiltrating from the outside or normal bacteria, leading to

the appearance of inflammatory ulcer reactions in the oral mucosa [4]. Therefore, accurate early diagnosis, prevention, and treatment of stomatitis are necessary for cancer patients receiving chemotherapy [4,5].

When worsened, the stomatitis, as such, can cause severe oral pain, lead to the necessity of fluid therapy due to reduced dietary intake, and cause increases in the hospitalization period and medical costs, resulting in economic burdens [5]. However, stomatitis is known to occur within 7 days after the end of chemotherapy, so that stomatitis symptoms cannot be easily found and, thus, can be easily overlooked by medical workers when the patent visits the medical institution for the next chemotherapy [6]. Therefore, accurate early diagnosis, prevention, and treatment of stomatitis are necessary for cancer patients receiving chemotherapy [7]. Meanwhile, *Momordica charantia Linn.* is a yearly Cucurbitaceae plant that lives in tropical regions, such as Asia, Amazon, and Africa, is used as a medicine [8], and contains various physiologically active ingredients, such as vitamin C, minerals, and beta-carotene. Flavan-3-o1 derivatives and phenolic acids have been reported as the main active ingredients of *Momordica charantia Linn.* [9]. *Momordica charantia Linn.* has been mainly used for the treatment of diabetes in Asia and Latin America, including Korea, and has been found to have antidiabetic effects thanks to the efficacy of charantin, which is known as a vegetable insulin, through in vitro experiments, animal experiments, and some human experiments [10,11]. In addition, the action of *Momordica charantia Linn.* as an antiviral and antitumor agent was found to be thanks to MAP-30, a 30-kD protein isolated from the seeds [12]. Furthermore, anticancer, antiulcer, analgesic, anti-inflammatory, and antioxidant effects have been reported [13–15].

*Mastic* is an extract of the sap extracted from the stem or leaves of the *Pistacia lentiscus* belonging to the family *Anacardiaceae* [16]. It has been used for gastrointestinal diseases and pain control [17], and has been reported to have anticancer effects. Studies on the anticancer effects of mastic include a study that reported the fact that mastic induces the apoptosis of colorectal cancer through a caspase-dependent pathway [18], and a study [19] that stated that mastic shows effects to inhibit prostate cancer through NF-kB gene regulation.

*Myrrh* (*Commiphora molmol*) is a medicine made by drying the milky fluid that flows when the bark of *Commiphora myrrha* belonging to the *Burseraceae* is damaged [20]. Its main ingredients are resin, essential oil, and rubber, and it also contains moisture and various impurities. The insoluble portion contains a-b-heerabomyrrholic acid, and the soluble portion contains a-b-heerabomyrrhol [21]. Since myrrh has antibacterial action, it has been used alone or in combination with other herbal medicines to treat infections and inflammation from old times, and it shows anti-inflammatory action [22]. The efficacy of myrrh's anticancer [21], antiphlogistic, anti-inflammatory [23], and antioxidant [24] effects have been reported, but no study findings of its effects on oral inflammation or on oral cancer have been reported yet. Although *Momordica charantia Linn.*, *Pistacia lentiscus*, and *Commiphora myrrha* are known to have various physiological effects, such as antidiabetic, anticancer and anti-inflammatory, respectively, no study related to the preventive or therapeutic pharmaceutical performance of mixtures extracted from them on oral solid cancer, in particular, or inflammation or cancer in the oral mucosa has been identified yet. In addition, the above medicinal plants have been reported to have excellent anticancer and anti-inflammatory effects on the gastrointestinal system; so, if they act in a complex way on the oral mucosa, they are expected to be very effective against oral cancer or stomatitis. However, there have been no reports of its efficacy yet. Therefore, this study is intended to examine the preventive and therapeutic effects of hot water, pressurized, ethanol, and supercritical extracts of the mixture of *Momordica charantia Linn.*, *Pistacia lentiscus*, and *Commiphora myrrha* on oral inflammation or on oral cancer.

## 2. Materials and Methods

This study is intended to examine the preventive and therapeutic effects of the supercritical extracts of *Momordica charantia Linn.*, *Pistacia lentiscus*, and *Commiphora myrrha* on oral inflammation and oral cancer. *Momordica charantia Linn.*, *Pistacia lentiscus*, and

*Commiphora myrrha* supplied from Jibio Pharm Co., Ltd. were cleaned and dried with hot air at 70 °C for 24 h, and pulverized to a size of 0.32 to 0.50 mm.

To that end, supercritical extract (MPC-4) was prepared as an experimental sample, and hot water extract (MPC-1), pressurized extract (MPC-2), and ethanol extract (MPC-3) were prepared as comparison samples from a solid phase product made by mixing *Momordica charantia Linn.* with *Pistacia lentiscus* and *Commiphora myrrhain* in weight ratios of 200 g:100 g:100 g.

In particular, when the total amount did not satisfy the weight ratio of 200 g:100 g:100 g in the pre-experiment, the inhibition and killing ability of bacteria in the oral mucosa was lowered. In addition, the cell proliferation inhibitory ability, the apoptosis inducing effect, and the intracellular signaling pathway regulation effect were not exhibited at the same time. Therefore, in this study, the total amount was determined as 200 g:100 g:100 g, and the experiment was carried out.

To determine the efficacy of the extracts as such, mouth-related bacteria inhibition and killing tests, cell proliferation inhibition tests in oral cancer cells, apoptosis inducing effect tests, intracellular signaling pathway regulation tests, and safety tests were performed. The procedures and methods of such tests are as follows.

### 2.1. Preparation of a Mixture of Momordica charantia Linn., Pistacia lentiscus, and Commiphora myrrha

Four different extract samples were prepared according to the extraction methods.

First, the sample (MPC-1) extracted with the hot water extraction method was obtained from the mixture by adding distilled water 10 times the weight of the mixture, and extraction was performed at 90 °C for 8 h using an autoclave (ST-50G, Jeiotech, Korea). The sample (MPC-2) extracted with the pressure extraction method was obtained by adding distilled water 10 times the weight, and extraction was performed at 90 °C for 8 h using an autoclave (ST-50G, Jeiotech, Korea). The sample (MPC-3) extracted with the ethanol extraction method was obtained by adding ethanol 10 times the weight of the mixture, and connecting a reflux tube for 8 h at 90 °C. The sample (MPC-4) extracted with the supercritical extraction method was prepared by repeating four times; the process to supply a supercritical fluid to a supercritical extractor at a flow rate of about 200 mL/min for 2 h while maintaining 50 °C, 300 bar, so that the supercritical fluid comes into contact with the solid phase dried product that filled the supercritical extractor (SC-CO$_2$ EXTRACTION SYSTEM, Ilshinautoclave Co. Ltd., Korea) to extract the extract from the solid phase dried product.

The extracts obtained through the above methods were filtered with 0.45 μm membrane filters, and then concentrated in vacuum and at room temperature for 3 h to prepare samples (Table 1).

**Table 1.** Sample preparation methods according to extraction methods.

| Comparison Samples | | | Experimental Sample |
|---|---|---|---|
| **MPC-1** | **MPC-2** | **MPC-3** | **MPC-4** |
| Hot Water Extraction Method | Pressurized Extraction Method | Ethanol Extraction Method | Supercritical Extraction Method |

### 2.2. Cell Proliferation Inhibition Test in Oral Cancer Cells

To identify the cell proliferation inhibitory effect in oral cancer cells, MC3 cells, a human oral cancer cell line, and HN22 cells, another human oral cancer cell line, were received from the Classroom of Oral Pathology, Department of Dentistry, Seoul National University and treated with various concentrations of *Momordica charantia Linn.* extract, and the cell viability was checked. The concentrations of *Momordica charantia Linn.* extract (MPC-1 to MPC-4) were prepared as 0 μg/mL, 10 μg/mL, 20 μg/mL, 30 μg/mL, and 40 μg/mL, and the oral cancer cells were treated for 48 h [25].

In order to identify cell proliferation inhibition, the abovementioned MC3 cells and HN22 cells were seeded in 6-well plates under $2.3 \times 10^5$ /mL and $2.0 \times 10^5$ /mL conditions, respectively. Thereafter, when the cells filled the wells 50–60%, the solvent control group was treated with 0.1% DMSO (FBS, Sigma, St. Louis, PA, USA), the treatment group was dissolved into DMEM (DMEM, Sigma, USA) containing 5% FBS (FBS, Sigma, St. Louis, PA, USA), the tubes were with methanol extract for 48 h, and, thereafter, the numbers of cells were measured using a Neubauer's chamber (hemocytometer, Neubauer improved, Sigma, St. Louis, PA, USA). The cells were washed with phosphate buffered saline (PBS, Sigma, USA), and after being treated with trypsin (Trypsin, Sigma, St. Louis, PA, USA), the cells were removed by centrifugation at $800 \times g$ rpm for 3 min. After adding 1 mL of PBS to the remaining cells, the contents were mixed with a pipette, and the total number of cells was counted using a hemocytometer. The above test was repeated three times, and the average value was measured.

### 2.3. Flow Cytometry Experiment

Flow cytometry is an experiment that can identify cells in which apoptosis is induced [26]. For flow cytometry, the FITC Annexin V-Apoptosis Detection Kit (BD Pharmingen, Franklin Lakes, NJ, USA) was used, and apoptosis was confirmed in the control (PBS) and extracts (MPC-1–4) of MC-3 oral cancer cells. In total, 40 µg/mL of the extract (MPC-1–4) and the control were treated with MC-3 cells and cultured for 24 h, and then the sample was pelleted through washing and centrifugation. Then, 3 µL of Annexin V-FITC and 1 µL of PI were treated, reacted in the dark for 15 min, and transferred to a FACS tube for flow cytometry analysis using a flow cytometer.

### 2.4. Intracellular Signaling Pathway Regulation Test

Representative factors related to cell death include apoptosis receptors (Fas/AP01, TNFR1), caspase family, p53, BCL-2 family and glutamate. For *Momordica charantia Linn.* extract (MPC-1~MPC-4), cell death was identified by analyzing caspase3, which plays an essential role in cell death signal transmission, and Bak, an apoptosis regulatory gene.

MC3 cells and HN22 cells prepared in the cell proliferation inhibition test in oral cancer cells were dispensed into 60 $mm^2$ and 10 $cm^2$ dishes, and DMSO (MDSO, Sigma, St. Louis, PA, USA) and test substances were treated by concentration. The relevant cells were quantified using lysis buffer (Lysis buffer, Sigma, St. Louis, PA, USA) and the DC protein assay [27]. Protein experimental materials were separated by electrophoresis on SDS polyacrylamide gel (EzWay$^{TM}$, Koma biotech, Korea) and transferred using PVDF membrane (PVDF membrane, Koma biotech, Korea). The abovementioned membrane was fixed in 5% skimmed milk at room temperature for 90 min, and primary antibodies Bak, Bid, and actin were incubated for 24 h at 4 °C. After washing with TBST (TBST, Sigma, St. Louis, PA, USA), the secondary antibody bound to HRP was incubated at room temperature for 90 min. The protein bound to the antibody was detected using an ECL solution (ECL solution, Sigma, St. Louis, PA, USA).

### 2.5. Safety Test

MTT assay, a representative cytotoxicity test, was used to evaluate the safety of MPC-1–4, and the quantification was carried out by modifying the Mosmann method. To that end, B16-F10 mouse melanoma cells (Korea Cell Line Bank) were dispensed at $1 \times 10^4$ cells/mL and incubated for 24 h. Thereafter, the medium was replaced with a new medium containing the extract samples diluted to concentrations of 100–1000 µg/mL, and the cells were incubated again for 24 h. Then, 20 µL of EZ-Cytox was added to each well, the cells were incubated for 1 h in a 5% $CO_2$ incubator at 37 °C, and the absorbance was measured thereafter with an ELISA reader (EpochTM 2, BioTek, Winooski, VT, USA). The cell viability calculation formula is as follows.

$$\text{Cell viability (\%)} = [(\text{Exp.} - \text{Blank}) / \text{Control}] \times 100 \qquad (1)$$

where: Exp, absorbance of extract including cells; Blank, absorbance of extract without cells; and Control, absorbance of distilled water containing cells.

### 2.6. Tumor Xenograft Model

An animal model was prepared to confirm the inhibitory effect of HN22 oral cancer cells in vivo. Six-week-old male nude mice(nu/nu) were obtained from Orient Bio Inc (Iksan, Korea). Each mouse was acclimatized for 1 week while supplying sufficient feed and water, and individually housed under regulated conditions (20–22 °C on a 12-h light/dark cycle with lights on at 7:00 AM). HN22 cells were subcutaneously injected ($5 \times 10^5$ cells/200 μL) into the back of the mice, and the mice were assigned randomly treatment groups [28]. The tumor volume was measured daily for 3 weeks in mice using vernier calipers (Mitutoyo, Japan). Fifteen mice were divided into the control group (water), low dose (MPC-4, 50 mg/kg), and high dose (MPC-4, 100 mg/kg) groups; five mice per group were orally administered for 21 days, and the weight was measured daily. Weight change was confirmed by measuring. After 3 weeks, mice in all groups were sacrificed with $CO_2$. Then, the tumor tissue was collected and weighed. After that, the liver, kidney, and spleen were removed by laparotomy, washed with saline to remove moisture with a filter paper, and then weighed.

### 2.7. Statistics and Data Processing

All experiments in this study were used for analysis based on the results of three or more independent runs under the same conditions, and all experimental results were expressed as Mean± Standard Deviation. After calculating the mean and standard deviation of the experimental results, statistical significance was verified by student's t-test.

## 3. Results

### 3.1. Cell Proliferation Inhibition Test in Oral Cancer Cells

In the case of MPC-4, the survival rate of oral cancer cells was analyzed to be the lowest. Concretely, MPC-4 reduced the cell proliferation concentration-dependently in MC3 and HN22 cells, which are oral cancer cells, and the survival rates of MC3 and HN22 cells at a concentration of 40 μg/mL were identified to be about 50%. In the case of MPC-1 and MPC-2, the survival rates of MC3 and HN22 cells were shown to be about 80%, which were consistent with the tendency of the minimum inhibitory concentration test. In the case of MPC-3, the survival rate of MC3 and HN22 cells was identified to be about 70% at a concentration of 40 μg/mL, indicating a lower oral cancer cell survival rate than that of MPC-4 (see Figure 1).

### 3.2. Results of Flow Cytometry Experiments

As a result of the flow cytometry experiment, it was confirmed that most of the MC3 cells survived with a survival rate of 96.6% in the control group (see Figure 2). In MPC-4, apoptosis induction was the most effective, and early apoptosis and late apoptosis were confirmed in 26.9% and 18.1%, respectively. Furthermore, the survival rate of oral cancer cells was decreased in MPC-1, MPC-2, and MPC-3 compared to the control group, but it was confirmed that it was not as effective as MPC-4.

### 3.3. Intracellular Signaling Pathway Regulation Experiment

Figure 3 shows the results of the intracellular signaling pathway regulation experiments by concentration. As can be seen in Figure 3, in the case of MPC-4, as a result of protein analysis to check the regulation of the signaling pathway, the expression of cleaved caspase 3 and Bak regulatory genes was found to have increased concentration-dependently in MC3 cells and HN22 cells, which are oral cancer cells (see Figure 4). On the other hand, in the case of MPC-1, MPC-2, and MPC-3, although the relevant genes increased concentration-dependently, the increases did not show values as significant as the value in the case of MPC-4.

### 3.4. Safety Experiment

As a result of the safety experiments conducted in this study, with regard to the cytotoxicity of the extracts in the cells, the cytotoxicity of MPC-1–4 was measured to be at least 95% on the basis of the cell viability of the untreated group (100%) at concentrations of 100 μg/mL, 300 μg/mL, 500 μg/mL, 700 μg/mL, and 1000 μg/mL, but no cytotoxicity appeared at all in other extracts at concentrations ranging from 0 to 1000 μg/mL.

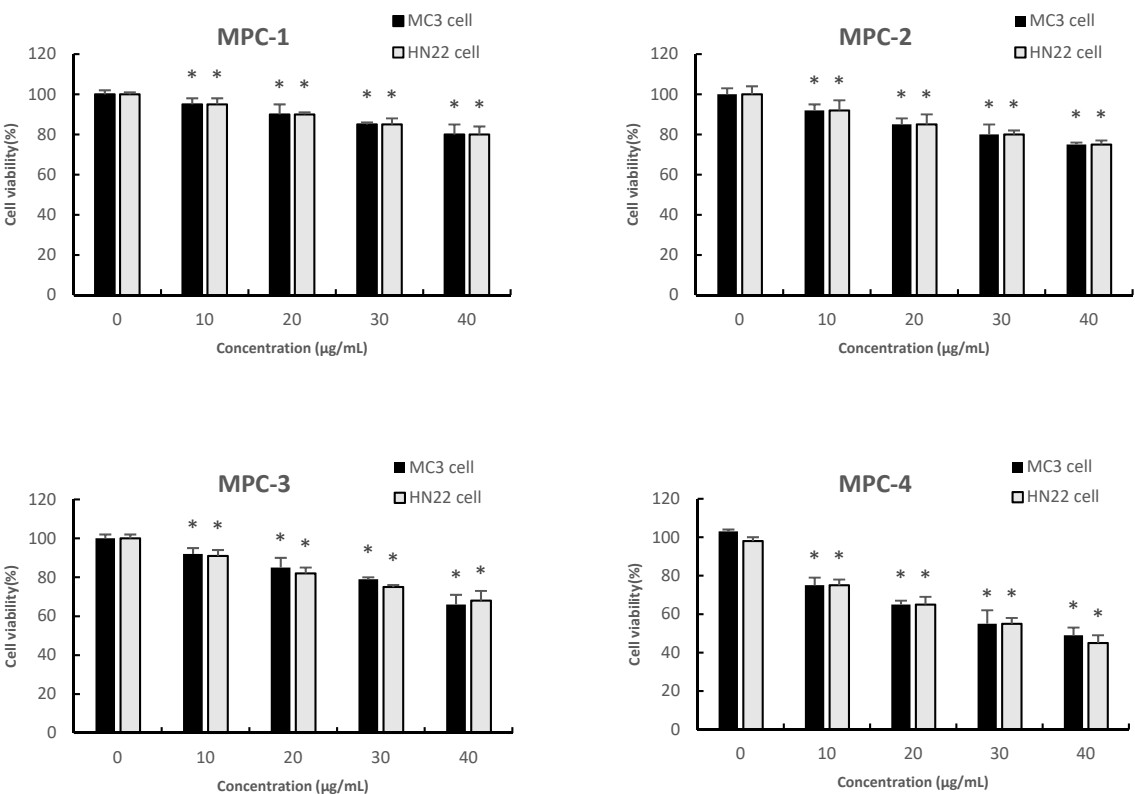

**Figure 1.** Oral cancer cell (MC3 and HN22 cell) survival rates by concentration. Statistical significance was determined by Student's *t*-test * $p < 0.05$).

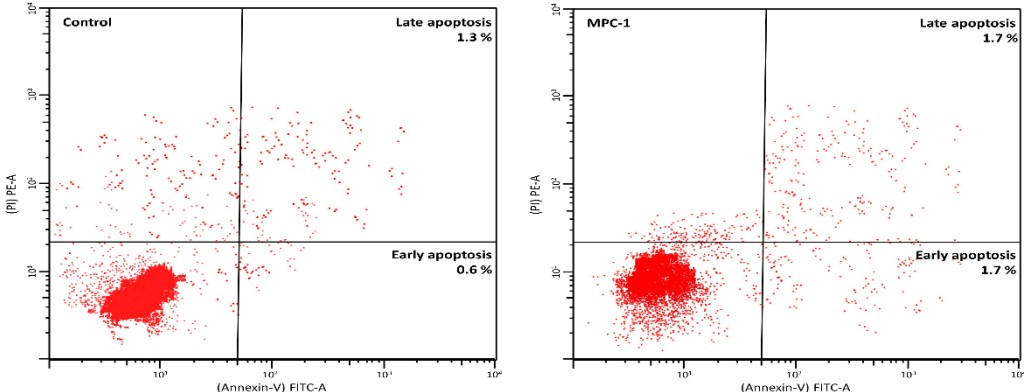

**Figure 2.** *Cont.*

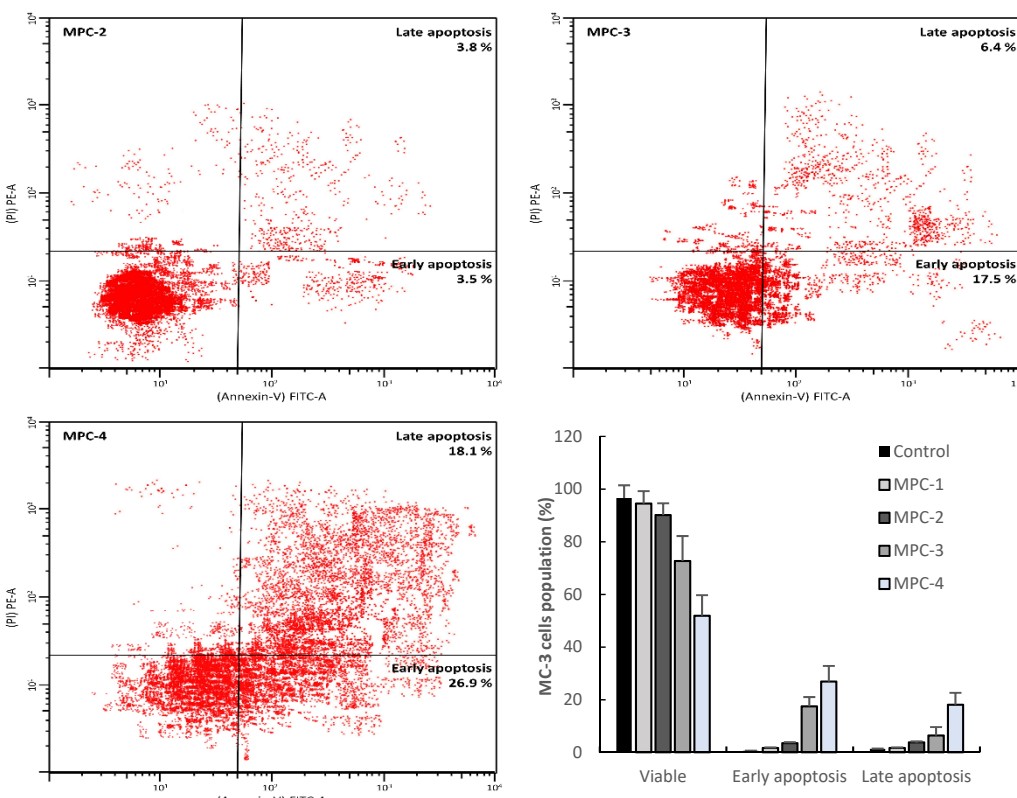

**Figure 2.** FACS analysis of oral cancer cells. MC3 cell were treated with 0, 10, 20, 30, and 40 μg/mL MPC-1–4 for 24 h and applied to flow cytometer after stain with FITC Annexin V and PI.

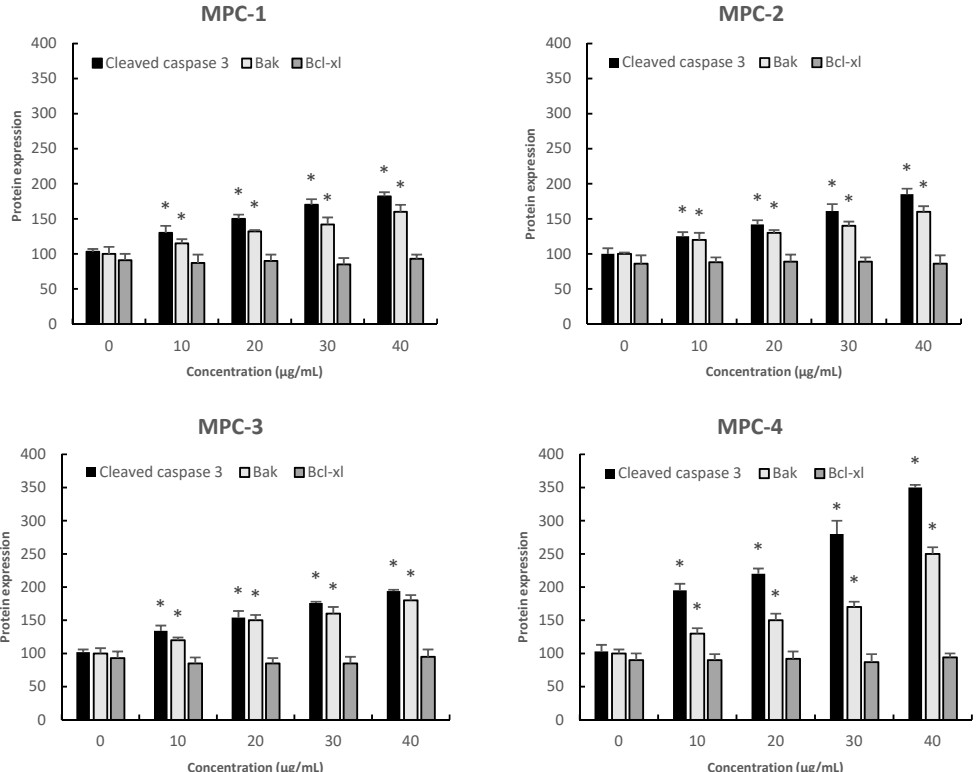

**Figure 3.** FACS analysis of oral cancer cells. MC3 cell were treated with 0, 10, 20, 30, and 40 μg/mL MPC-1–4 for 24 h and applied to flow cytometer after stain with FITC Annexin V and PI. Statistical significance was determined by Student's *t*-test (* $p < 0.05$).

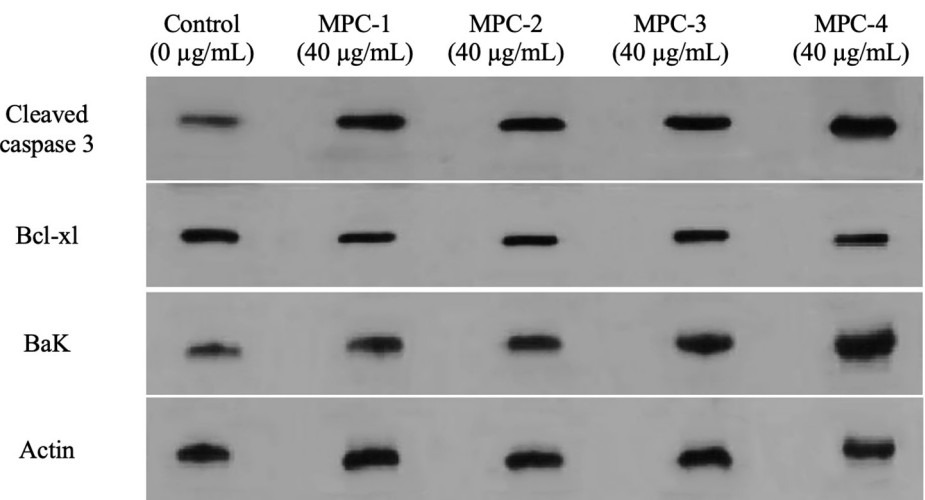

**Figure 4.** Protein expression effect.

### 3.5. Tumor Volume and Weight in Xenograft Model

As a result of the tumor volume of nude mice inoculated with HN22, tumor growth inhibition was confirmed in the group administered with MPC-4 (high dose [100 mg/kg], low dose [50 mg/kg]) compared to the control group administered with water. Moreover, there was no significant difference in weight change (Figure 5). It was confirmed that the volume of the tumor compared to the control group was suppressed by 59.6% in the high-dose (100 mg/kg) group and 39.0% in the low-dose (50 mg/kg) group. It was confirmed that tumor growth was inhibited by 54.2% in the high-dose (100 mg/kg) group and 36.1% in the low-dose (50 mg/kg) group (Table 2).

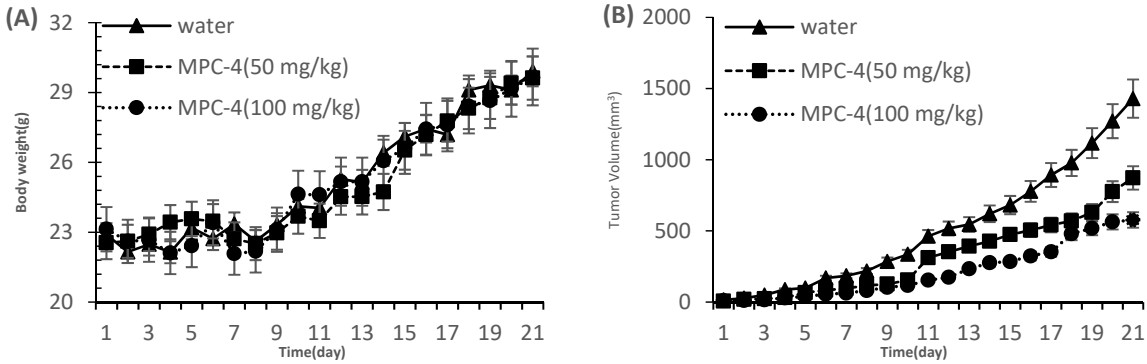

**Figure 5.** Changes in body weight (**A**) and tumor volume (**B**) in Xenograft models.

**Table 2.** Tumor inhibited rate in Xenograft model.

| MPC-4 (mg/kg) | Tumor Volume (mm³) | | | Tumor Weight (g) | |
|---|---|---|---|---|---|
| | Pre-Experiment | Post-Experiment | Inhibited Rate (%) | Post-Experiment | Inhibited Rate (%) |
| 0 | $14.7 \pm 1.1$ | $1429.1 \pm 134.5$ | - | $0.66 \pm 0.31$ | - |
| 50 | $6.7 \pm 0.9$ | $872.1 \pm 81.9$ | 39.0 | $0.42 \pm 0.25$ | 36.1 |
| 100 | $10.1 \pm 0.9$ | $576.8 \pm 53.4$ | 59.6 | $0.30 \pm 0.16$ | 54.2 |

### 3.6. Organ Weight Measurement in Xenograft Model

In order to determine the in vivo toxicity of MPC-4 ingestion, the liver, kidney, and spleen of sacrificed mice were collected and the weights of the MPC-4 group (high-dose,

low-dose) and the control group were compared. As a result, the high-dose (100 mg/kg) group and the low-dose (50 mg/kg) group administered orally with MPC-4 did not show any difference in organ weight compared to the control group fed with water. It is predicted that it does not have any toxicity (Figure 6).

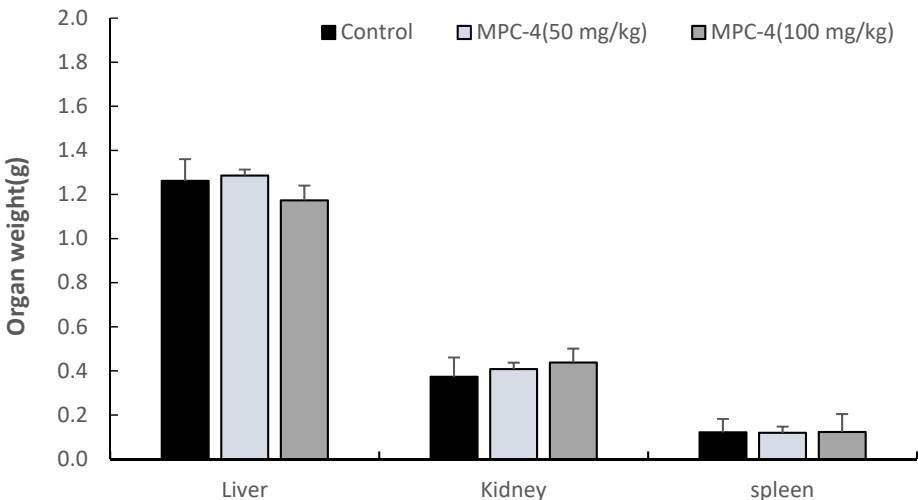

**Figure 6.** Weight by organ (Liver, Kidney, Spleen) in Xenograft models. Statistical significance was determined by Student's *t*-test.

## 4. Discussion and Conclusions

Unlike normal cells, abnormally growing cancer cells have characteristics such as hyperproliferation and metastasis [29]. In order to suppress this occurrence, it is important to induce anticancer activity by inducing apoptosis while reducing the destruction of normal cells. Recently, various treatment methods have been researched to kill cancer cells, and, in particular, many methods using natural substances are being developed. Therefore, in this study, samples (MPC-1–4) were prepared using natural substances *Momordica charantia Linn., Pistacia lentiscus,* and *Commiphora myrrha* according to the extraction method, and the efficacy of these samples for treating inflammation in the oral cavity and oral cancer was investigated.

MPC-4 extracted by the supercritical method decreased the proliferation of oral cancer cells, MC3, and HN22 cells, in a concentration-dependent manner ($p < 0.01$), and at a concentration of 40 μg/mL, the survival rates of MC3 and HN22 cells were 49% and 45%, respectively. On the other hand, MPC-1–3 samples also decreased the survival rate of oral cancer cells in a concentration-dependent manner ($p < 0.01$). However, as the survival rates were 80%, 75%, and 67%, respectively, the oral cancer cell proliferation inhibitory ability was lower than that of MPC-4. These results suggest that the extraction method may affect the inhibition of MC3 and HN22 cell proliferation.

Supercritical Fluid Extraction has excellent selectivity, such as fractionation and separation, so a high-purity product can be obtained, and the extraction solvent can be recovered almost completely without loss, thus obtaining a purified product without residual solvent [30]. In addition, due to its low viscosity, it has good permeability to the sample, so the extraction efficiency is high, and the extraction rate is fast with a high diffusion coefficient, and damage to nutrients due to heat can be avoided by low-temperature extraction. In particular, in this study, the difference in density between a mixture such as bitter melon and the supercritical fluid was large and the viscosity of the supercritical fluid was low, so it was easy to separate the solvent from the mixture. As a result, it is considered that the active ingredient that inhibits oral cancer cell proliferation could be more efficiently extracted.

During apoptosis, caspase accelerates cell death through proteolysis of over 400 proteins. Caspase is activated through the intrinsic cell death pathway [31]. The intrinsic cell death pathway is driven by the Bcl-2 protein family, which regulates apoptosis via

mitochondria. At this time, the outer mitochondrial membrane is permeabilized and cells begin to die. Proteins released from the intermembrane space of mitochondria promote caspase activation and apoptosis [32]. The released cytochrome C binds to APAF-1 and induces caspase 9 activity, which in turn activates caspase 3 and 7, leading to apoptosis. As a result of flow cytometry, early apoptosis and late apoptosis in the control group were 0.6% and 1.3%, respectively, whereas in MPC-4, it was 26.9% and 18.1%, indicating that apoptosis induction was the most effective. MPC-1–2 showed low induction of apoptosis, and MPC-3 showed a higher induction effect than MPC-1–2, but lower than MPC-4. Therefore, it was confirmed that the extract induces apoptosis in MC3 cells.

In addition, as a result of protein analysis to confirm the regulation of signaling pathways, expression of cleaved caspase 3 and Bak regulatory genes increased in a concentration-dependent manner in MC3 and HN22 cells ($p < 0.01$). Cleaved caspase 3 is a factor that plays a core role in inducing apoptosis and is known to exist in an inactive state in the intracellular nuclei and the outer membrane of mitochondria until it is activated by stimuli that induce apoptosis [31]. It is judged that MPC-4 induces apoptosis by Bak in MC3 cells and induces the truncation of Bid to induce apoptosis in HN22 cells. Meanwhile, a study conducted by Hwang (2018) verified effects against glioma [33], a type of cancer, based on the antioxidant efficacy of *Momordica charantia Linn.* extract. Glioma is a malignant tumor that occurs in glial cells (neuroglia) or precursor cells of the brain [34], and it was identified that *Momordica charantia Linn.* extract upregulated the antioxidant activity of glioma cancer cells and reduced the intracellular active oxygen level. That is, *Momordica charantia Linn.* extract was found to inhibit the cancer cells survival rate and proliferation of glioma cells, stop the cell cycle, induce cancer cell death, and increase the protein expression level.

As a result of the safety test, MPC-4 was measured to be greater than 95% at a concentration of 100 to 1000 μg/mL based on the cell viability (cell viability: 100%) of the untreated group. This means that there is no cytotoxicity of the MPC-4 sample. In addition, as a result of comparing the weights of the liver, kidney, and spleen of the experimental group and the control group to determine the presence or absence of toxicity in vivo, there was no statistically significant difference. Therefore, it is judged that MPC-4 does not have toxicity to oral administration. According to the results of the Xenograft model, which measured the volume and weight of the tumor, MPC-4 inhibited tumor growth in the high-dose (100 mg/kg) group, rather than the low-dose (50 mg/kg) group. This means that MPC-4 more effectively inhibits oral cancer cell proliferation and induces apoptosis in the high-dose (100 mg/kg) group.

In summary, MPC-4 showed the best results in all aspects, such as inhibition of cell proliferation, induction of apoptosis, regulation of intracellular signaling pathways, safety, and Xenograft model in oral cancer cells. These results suggest that the supercritical extract of *Momordica charantia Linn., Pistacia lentiscus*, and *Commiphora myrrha* can be applied as a preventive and therapeutic agent for inflammation and carcinoma of the oral mucosa in the future.

**Author Contributions:** M.J.C. conceived and designed the study; H.S.Y. and S.J.P. finished the revision of the manuscript. The manuscript was produced through the contributions of all authors. All authors have read and agreed to the published version of the manuscript.

**Funding:** This research received no external funding.

**Institutional Review Board Statement:** Not applicable.

**Informed Consent Statement:** Not applicable.

**Data Availability Statement:** Not applicable.

**Conflicts of Interest:** The authors declare no conflict of interest.

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
