# Peer review of "The Effects of the Supercritical Extracts of Momordica charantia Linn., Pistacia lentiscus, and Commiphora myrrha on Oral Inflammation and Oral Cancer"

_sustainability, doi:10.3390/su14042458_

Round 1

Reviewer 1 Report

In this study, the author investigated the potential protective effects of extracts of Momordica charantia, Pistacia lentiscus, and Commiphora myrrha against oral inflammation and cancer. Results from the following analyses were presented: MIC assay on mouth-related bacteria, cancer cell proliferation assay, apoptosis assay, protein expression analysis on two cell death regulators, cytotoxicity test, and tumor xenograft.

Overall, the scope appeared wide enough.

However, I found some major issues which must be addressed adequately before the manuscript can be considered any further.

  1. Controls are required in at least some of the analyses.
  2. Statistical analyses are required for at least some of the results.
  3. Data presented should be interpreted sufficiently in DISCUSSION. The current DISCUSSION largely looks like a repetition/re-descriptions of RESULTS and the interpretation appears superficial/too limited. The significance/meaning of the results presented seems not carefully thought through.
  4. The author mentioned that the study aimed at investigating the protective effects of the samples against oral inflammation. Please see INTRODUCTION - last statement “Therefore, this study is intended to examine the preventive and therapeutic effects … on oral inflammation or on oral cancer.” However, no data/result was presented at all on the anti-inflammation effect of the samples in the whole paper. Also, there was no mention about anti-inflammation at all in the concluding paragraph.
  5. The overall writing is confusing/incoherent and appears to be not well-checked before submission; this should be improved.
  6. Importantly, could the author please do not directly copy statements word-by-word from the main text to the ABSTRACT, or from one section to another section in the main text? Please see specific comments below for examples where the same statements – word-by-word – are found in different sections of the manuscript. This practice is unacceptable. Please make an extra effort to rephrase them.

Below are my specific comments:

  1. Problems in scientific names
  • All scientific names – whether they are the species of the plant samples or those of the bacteria used – MUST BE in italics.
  • “Momordica charantia linn” – The standard/acceptable way to write “linn” is “” Please note the uppercase “L” and the period at the end.
  • Please correct the above - where appropriate - in the whole manuscript.

  1. ABSTRACT
  • Diseases in the oral cavity cause fatal pain to humans from a sustainable point of view.” – Please recheck or revise this. It is unclear how it can be so “from a sustainable point of view.”
  • In its current form, it seems too vague/general. It should be revised to present key findings more specially/explicitly/quantitatively.
  • ABSTRACT also contains redundant/repetitive information. For example – please rewrite these statements to make them more concise: “To experiment the efficacy of the extracts as such, mouth-related bacteria inhibition and killing tests, cell proliferation inhibition tests in oral cancer cells, apoptosis inducing effect tests, intracellular signaling pathway regulation tests, and safety tests were performed. According to the results of this study, compared to the hot water, pressurized, and ethanol extraction methods, the supercritical extraction method showed the best results in all aspects of mouth-related bacterial inhibition and killing, cell proliferation inhibition in oral cancer cells, apoptosis induction, intracellular signaling pathway regulation, and safety.
  • The last statement “In this study, through …” MUST BE revised. It gave the wrong impression that this study has presented results on antioxidant, antidiabetic and anti-inflammation effects. But those effects were actually reported by others. In the concluding remark in the ABSTRACT, could the author please write a conclusion based on the actual findings presented in this study?

  1. INTRODUCTION
  • Page 1 – “In particular, oral disease is a very important part of the body for humans in that it is a pathway for humans to consume nutrients necessary to maintain a sustainable life.” – Please change “oral disease” to “the mouth” or “the oral cavity”.
  • The writing is difficult to read because it is a single, very long paragraph. It would be helpful to readers if the author could split the paragraph into shorter paragraphs.
  • A more specific/convincing justification should be given as to why the author decided to work on this particular combination of these three plant species. There are also other plant species with similar health benefits/potential as one or all of the three species. So please explain at least briefly why this particular three were chosen.

  1. MATERIALS AND METHODS – paragraph 1
  • To that end, Momordica charantia linn supercritical extract (MPC-4) was prepared … in weight ratios of 1:0.5~1.5.”. – This statement should be rewritten for the following reasons:
    • It is the same statement – word-by-word – as that appears in the ABSTRACT. Please make an extra effort to rephrase the statement in either the ABSTRACT or in M&M.
    • The statement is just too long and difficult to read. Please split it into two statements if possible.
    • The statement is misleading/confusing. If the samples each consisted of three species, then it would be misleading/confusing to name them extracts of “Momordica charantia linn”.
    • The statement mentioned “weight ratios of 1:0.5~1.5.” – is that a typo error? Considering there were three herb species involved, it seems more likely to be 1: 0.5:1.5. Please recheck.
  • Importantly, the author should indicate in M&M – or in other parts of the paper – the rationale of choosing the “weight ratios of 1:0.5~1.5.” Why not other weight ratios? Was this decision based on any previous studies/optimization work?
  • One more issue - there seems to be hardly any cited references in M&M. Can the author recheck to see whether this is appropriate, or were they missed out?

  1. MATERIALS AND METHODS – paragraph 2
  • To experiment the efficacy of the extracts … safety tests were performed.” – It is the same statement – word-by-word – as that appears in the ABSTRACT. Please make an extra effort to rephrase the statement in either the ABSTRACT or in M&M.

  1. MATERIALS AND METHODS – Section 2.1
  • Paragraph 1 – “… were dried at 70°C for 24 hours and pulverized into 0.32~0.50 mm sizes …” – This information was already provided in paragraph 1 (the last line of page 2). The information is repetitive/redundant. Please revise.
  • Paragraphs 2 & 3 – Please indicate how the extraction conditions/parameters were established. Please cite references where appropriate. Also, please rewrite the statements. They are too long that it is challenging to read and understand them immediately without difficulty.

  1. MATERIALS AND METHODS – Section 2.2
  • Paragraph 1 – “… sobrinus 6715, S. mutans GS5, …E. faecalis ATCC 4083 … P. gingivalis 2561, P. intermedia ATCC 25611 …” – Please write the species names in italics. Also, at the first mention, the genus should be written in full.
  • It is unclear why if the focus of the study was on protection against oral inflammation / cancer, the study should begin by testing for antibacterial activity. Justification/explanation is required (at least briefly) in M&M or in INTRODUCTION.
  • Paragraph 2 – “The experimental group was … a liquid medium ...” – Please replace “The experimental group was” with “The samples were”. Please specify what “liquid medium” refers to.
  • Did the author refer to any papers/references for the methodology? If yes, please cite references where appropriate.

  1. MATERIALS AND METHODS – Section 2.3
  • Paragraph 1 – “… treated for 48 hours” – Is that any evidence/support that indicates that this is the optimum/appropriate treatment duration? Please clarify.
  • Did the author refer to any papers/references for the methodology? If yes, please cite references where appropriate.

  1. MATERIALS AND METHODS – Sections 2.4-2.7
  • Did the author refer to any papers/references for the methodology? If yes, please cite references where appropriate.

  1. MATERIALS AND METHODS – Section 2.6
  • For this part, it makes more sense to test for safety/cytotoxicity on a non-cancerous cell line, rather than on a melanoma cell line. Can the author please explain why the test was performed on a cancerous cell line instead?
  • Why was the test done by using MPC-1, but not MPC-4? Please recheck.

  1. MATERIALS AND METHODS – Please also add a section to describe the experimental design, e.g., number of replications, statistical tests, statistical software used, whether the error bars refer to standard deviation or standard errors.

  1. RESULTS – Sections 3.1 - 3.4
  • Controls are missing. For Table 2 – the test should include one or more appropriate antibiotics/antibacterial agents to serve as controls. Similarly, for Figures 1 and 2, the test should include one or more appropriate anticancer drugs/compounds to serve as controls.
  • For Table 2 and Figure 1 - 3 – please perform statistical tests on the results. Also, please indicate the number of replicates done.
  • For Figure 2 – Can the author please include images/photographs of apoptotic nuclear fragmentation for at least the 0 and 40 ug/mL MPC-4 treatments - as evidence?
  • For Figure 3 – Can the author please include images/photographs of the blots as evidence?

  1. RESULTS – Section 3.6
  • Can the author please include images/photographs of the tumor growth in the animal model as evidence?
  • For Table 3 – owing to the large standard errors/deviation, the differences perceived for tumor weight might not be statistically significant/meaningful. To resolve this, please perform statistical tests on the results and reinterpret the data. Also, please indicate the number of replicates done.

  1. RESULTS – Section 3.7
  • Please indicate the number of replicates done.

  1. DISCUSSION and CONCLUSIONS
  • Overall, this part seems poorly written, lacking coherence, and lacking discussion in relation to previous observations in the literature. Interpretation of results seems superficial. Please improve this.
  • More importantly, it largely looks like RESULTS – as most of the text simply repeated information already described in RESULTS. In some cases, the repeated statements were simply word-by-word repetition, which is not acceptable. Please improve this.
  • Paragraph 2 - “… in this study the efficacy of Momordica charantia linn fruits, … was experimentally investigated.” – This statement appears TWO times – in ABSTRACT and DISCUSSION – word-by-word. Please rephrase.
  • Paragraph 2 – “To that end, Momordica charantia linn supercritical extract (MPC-4) was … safety tests were performed.” – These two statements appear THREE times - in ABSTRACT, M&M, and DISCUSSION – word-by-word. Please amend.
  • Paragraphs beginning with the following: “First, as a result of the mouth-related bacteria inhibition and killing tests,…”, “Second, as a result of the cell proliferation inhibition experiment…”, “Third, as a result of the apoptosis-inducing effect experiment,…”, “Fourth, as a result of the intracellular signal transduction pathway regulation experiment,…”, “Fifth, as a result of the safety test, with regard to the cytotoxicity of the extracts in the cells,…”, and “Sixth, as a result of the animal model experiment, …” – these paragraphs are all redundant and repeated descriptions of information already provided in RESULTS. Please amend them.
  • Paragraph 10 - The statement “In this study, through the supercritical extraction method, the fact that … was presented.” - This statement appears TWO times – in ABSTRACT and DISCUSSION – word-by-word. Please rephrase.
  • Paragraph 11 – “Hot water Momordica charantia linn extract, pressurized Momordica charantia linn…, and intracellular signaling pathways regulating effects” – These are redundant and repeated descriptions of information already provided in RESULTS. No proper discussion/interpretation of the significance/meaning of results was presented.
  • Paragraph 13 – “expression of cleaved caspase 3 and Bak regulatory genes” – Please be careful to indicate that what was presented was protein expression results. Not gene expression results.

Author Response

Response to Reviewer 1 Comments

In this study, the author investigated the potential protective effects of extracts of Momordica charantiaPistacia lentiscus, and Commiphora myrrha against oral inflammation and cancer. Results from the following analyses were presented: MIC assay on mouth-related bacteria, cancer cell proliferation assay, apoptosis assay, protein expression analysis on two cell death regulators, cytotoxicity test, and tumor xenograft.

Overall, the scope appeared wide enough.

However, I found some major issues which must be addressed adequately before the manuscript can be considered any further.

  1. Controls are required in at least some of the analyses.
  2. Statistical analyses are required for at least some of the results.
  3. Data presented should be interpreted sufficiently in DISCUSSION. The current DISCUSSION largely looks like a repetition/re-descriptions of RESULTS and the interpretation appears superficial/too limited. The significance/meaning of the results presented seems not carefully thought through.
  4. The author mentioned that the study aimed at investigating the protective effects of the samples against oral inflammation. Please see INTRODUCTION - last statement “Therefore, this study is intended to examine the preventive and therapeutic effects … on oral inflammation or on oral cancer.” However, no data/result was presented at all on the anti-inflammation effect of the samples in the whole paper. Also, there was no mention about anti-inflammation at all in the concluding paragraph.
  5. The overall writing is confusing/incoherent and appears to be not well-checked before submission; this should be improved.
  6. Importantly, could the author please do not directly copy statements word-by-word from the main text to the ABSTRACT, or from one section to another section in the main text? Please see specific comments below for examples where the same statements – word-by-word – are found in different sections of the manuscript. This practice is unacceptable. Please make an extra effort to rephrase them.

Below are my specific comments:

  1. Problems in scientific names
  • All scientific names – whether they are the species of the plant samples or those of the bacteria used – MUST BE in italics.
  • “Momordica charantia linn” – The standard/acceptable way to write “linn” is “” Please note the uppercase “L” and the period at the end.
  • Please correct the above - where appropriate - in the whole manuscript.

All scientific names have been changed to italics. (linn -> Linn.)

  1. ABSTRACT
  • Diseases in the oral cavity cause fatal pain to humans from a sustainable point of view.” – Please recheck or revise this. It is unclear how it can be so “from a sustainable point of view.”
  • In its current form, it seems too vague/general. It should be revised to present key findings more specially/explicitly/quantitatively.
  • ABSTRACT also contains redundant/repetitive information. For example – please rewrite these statements to make them more concise: “To experiment the efficacy of the extracts as such, mouth-related bacteria inhibition and killing tests, cell proliferation inhibition tests in oral cancer cells, apoptosis inducing effect tests, intracellular signaling pathway regulation tests, and safety tests were performed. According to the results of this study, compared to the hot water, pressurized, and ethanol extraction methods, the supercritical extraction method showed the best results in all aspects of mouth-related bacterial inhibition and killing, cell proliferation inhibition in oral cancer cells, apoptosis induction, intracellular signaling pathway regulation, and safety.
  • The last statement “In this study, through …” MUST BE revised. It gave the wrong impression that this study has presented results on antioxidant, antidiabetic and anti-inflammation effects. But those effects were actually reported by others. In the concluding remark in the ABSTRACT, could the author please write a conclusion based on the actual findings presented in this study?

All abstracts have been rewritten according to the review comments. Duplication is minimized and only the main content is described.

  1. INTRODUCTION
  • Page 1 – “In particular, oral disease is a very important part of the body for humans in that it is a pathway for humans to consume nutrients necessary to maintain a sustainable life.” – Please change “oral disease” to “the mouth” or “the oral cavity”.

 “oral disease” has been changed to “the oral cavity”.

  • The writing is difficult to read because it is a single, very long paragraph. It would be helpful to readers if the author could split the paragraph into shorter paragraphs.

Long sentences have been rewritten.

  • A more specific/convincing justification should be given as to why the author decided to work on this particular combination of these three plant species. There are also other plant species with similar health benefits/potential as one or all of the three species. So please explain at least briefly why this particular three were chosen.

The following information has been inserted to present the justification for the three medicinal plants selected in this study.

 (In addition, the above medicinal plants have been reported to have excellent anticancer and anti-inflammatory effects on the gastrointestinal system, so if they act in a complex way on the oral mucosa, they are expected to be very effective against oral cancer or stomatitis. However, there have been no reports of its efficacy yet.)

  1. MATERIALS AND METHODS – paragraph 1
  • To that end, Momordica charantia linn supercritical extract (MPC-4) was prepared … in weight ratios of 1:0.5~1.5.”. – This statement should be rewritten for the following reasons:
    • It is the same statement – word-by-word – as that appears in the ABSTRACT. Please make an extra effort to rephrase the statement in either the ABSTRACT or in M&M.

The abstract has been rewritten so that it does not overlap with the main text.

    • The statement is just too long and difficult to read. Please split it into two statements if possible.

Long sentences were made as simple and clear as possible.

    • The statement is misleading/confusing. If the samples each consisted of three species, then it would be misleading/confusing to name them extracts of “Momordica charantia linn”.
    • The statement mentioned “weight ratios of 1:0.5~1.5.” – is that a typo error? Considering there were three herb species involved, it seems more likely to be 1: 0.5:1.5. Please recheck.

The weight ratio has been modified to 200g:100g:100g.

  • Importantly, the author should indicate in M&M – or in other parts of the paper – the rationale of choosing the “weight ratios of 1:0.5~1.5.” Why not other weight ratios? Was this decision based on any previous studies/optimization work?

To present the basis for the weight ratio, the pre-experimental results are specified as follows.

(“In particular, when the total amount did not satisfy the weight ratio of 200g:100g:100g in the pre-experiment, the inhibition and killing ability of bacteria in the oral mucosa was lowered. In addition, the cell proliferation inhibitory ability, the apoptosis inducing effect, and the intracellular signaling pathway regulation effect were not exhibited at the same time. Therefore, in this study, the total amount was determined as 200g:100g:100g and the experiment was carried out.”)

  • One more issue - there seems to be hardly any cited references in M&M. Can the author recheck to see whether this is appropriate, or were they missed out?

We have supplemented the four cited references in the Materials and Methods section.

[25] Jang, B., Oh, S. J., Shin, J., Lee, H. E., Jeon, J. G., & Cho, S. D. (2014). Effect of Methanol Extract of Dryopteris Crassirhizoma in Human Oral Cancer Cells. Journal of Food Hygiene and Safety, 29(3), 248-251.

[26] Shapiro, H. M. (2005). Practical flow cytometry. John Wiley & Sons.

[27] Lim, S. Y., Choi, J. K., Shin, S. J., Kwon, S. H., Cho, C. H., Cha, S. D., & Song, D. K. (2009). Induction of growth inhibition and apoptosis in human endometrial cancer cells by histone deacetylase inhibitors. Korean Journal of Obstetrics and Gynecology, 911-919.

[28] Kim, K. H., Kim, C. H., Han, S. J., & Lee, J. H. (2006). THE ANTICANCER EFFECT OF PACLITAXEL IN ORAL SQUAMOUS CELL CARCINOMA XENOGRAFT. Maxillofacial Plastic and Reconstructive Surgery, 28(2), 95-110.

  1. MATERIALS AND METHODS – paragraph 2
  • To experiment the efficacy of the extracts … safety tests were performed.” – It is the same statement – word-by-word – as that appears in the ABSTRACT. Please make an extra effort to rephrase the statement in either the ABSTRACT or in M&M.

 The abstract has been rewritten so that it does not overlap with the main text.

  1. MATERIALS AND METHODS – Section 2.1
  • Paragraph 1 – “… were dried at 70°C for 24 hours and pulverized into 0.32~0.50 mm sizes …” – This information was already provided in paragraph 1 (the last line of page 2). The information is repetitive/redundant. Please revise.

Repeated and duplicated content has been deleted.

  • Paragraphs 2 & 3 – Please indicate how the extraction conditions/parameters were established. Please cite references where appropriate. Also, please rewrite the statements. They are too long that it is challenging to read and understand them immediately without difficulty.

 The text has been edited to be concise.

  1. MATERIALS AND METHODS – Section 2.2
  • Paragraph 1 – “… sobrinus 6715, S. mutans GS5, …E. faecalis ATCC 4083 … P. gingivalis 2561, P. intermedia ATCC 25611 …” – Please write the species names in italics. Also, at the first mention, the genus should be written in full.

The species name is written in italics, and the first mentioned genus is written in its full name.

  • It is unclear why if the focus of the study was on protection against oral inflammation / cancer, the study should begin by testing for antibacterial activity. Justification/explanation is required (at least briefly) in M&M or in INTRODUCTION.

 It has been changed to an experimental item suitable for the purpose of this study.

(Before revision)

2.2. Mouth-related bacteria inhibition and killing tests were deleted.

2.3. Cell proliferation inhibition test in oral cancer cells

2.4. Apoptosis inducing effect test

2.5. Intracellular signaling pathway regulation test

2.6. safety test

2.7. Tumor Xenograft Model

(After modification)

2.2. Cell proliferation inhibition test in oral cancer cells

2.3. flow cytometry experiment

2.4. Intracellular signaling pathway regulation test

2.5. safety test

2.6. Tumor Xenograft Model

2.7. Statistics and data processing

  • Paragraph 2 – “The experimental group was … a liquid medium ...” – Please replace “The experimental group was” with “The samples were”. Please specify what “liquid medium” refers to.

Edited according to comments.

  • Did the author refer to any papers/references for the methodology? If yes, please cite references where appropriate.

The following references have been added.

[25] Jang, B., Oh, S. J., Shin, J., Lee, H. E., Jeon, J. G., & Cho, S. D. (2014). Effect of Methanol Extract of Dryopteris Crassirhizoma in Human Oral Cancer Cells. Journal of Food Hygiene and Safety, 29(3), 248-251.

  1. MATERIALS AND METHODS – Section 2.3
  • Paragraph 1 – “… treated for 48 hours” – Is that any evidence/support that indicates that this is the optimum/appropriate treatment duration? Please clarify.

Relevant previous studies based on 48 hours were cited.

[25] Jang, B., Oh, S. J., Shin, J., Lee, H. E., Jeon, J. G., & Cho, S. D. (2014). Effect of Methanol Extract of Dryopteris Crassirhizoma in Human Oral Cancer Cells. Journal of Food Hygiene and Safety, 29(3), 248-251.

  • Did the author refer to any papers/references for the methodology? If yes, please cite references where appropriate.

Relevant prior research has been cited and added to the bibliography.

  1. MATERIALS AND METHODS – Sections 2.4-2.7
  • Did the author refer to any papers/references for the methodology? If yes, please cite references where appropriate.

The following two related prior studies have been cited and added to the references.

[27] Lim, S. Y., Choi, J. K., Shin, S. J., Kwon, S. H., Cho, C. H., Cha, S. D., & Song, D. K. (2009). Induction of growth inhibition and apoptosis in human endometrial cancer cells by histone deacetylase inhibitors. Korean Journal of Obstetrics and Gynecology, 911-919.

[28] Kim, K. H., Kim, C. H., Han, S. J., & Lee, J. H. (2006). THE ANTICANCER EFFECT OF PACLITAXEL IN ORAL SQUAMOUS CELL CARCINOMA XENOGRAFT. Maxillofacial Plastic and Reconstructive Surgery, 28(2), 95-110.

  1. MATERIALS AND METHODS – Section 2.6
  • For this part, it makes more sense to test for safety/cytotoxicity on a non-cancerous cell line, rather than on a melanoma cell line. Can the author please explain why the test was performed on a cancerous cell line instead?

There is no special reason. However, if you give your opinion on this part of the experiment, we will supplement or delete it.

  • Why was the test done by using MPC-1, but not MPC-4? Please recheck.

It was incorrectly described as MPC-1, so it was changed to MPC-4.

  1. MATERIALS AND METHODS – Please also add a section to describe the experimental design, e.g., number of replications, statistical tests, statistical software used, whether the error bars refer to standard deviation or standard errors.

Statistics and data processing are described below.

2.7. Statistics and data processing

All experiments in this study were used for analysis based on the results of three or more independent runs under the same conditions, and all experimental results were expressed as Mean± Standard Deviation. After calculating the mean and standard deviation of the experimental results, statistical significance was verified by student’s t-test.

  1. RESULTS – Sections 3.1 - 3.4
  • Controls are missing. For Table 2 – the test should include one or more appropriate antibiotics/antibacterial agents to serve as controls. Similarly, for Figures 1 and 2, the test should include one or more appropriate anticancer drugs/compounds to serve as controls.

“2.2. Mouth-related bacteria inhibition and killing tests” were deleted.

  • For Table 2 and Figure 1 - 3 – please perform statistical tests on the results. Also, please indicate the number of replicates done.

A statistical analysis result (t-test) has been added.

  • For Figure 2 – Can the author please include images/photographs of apoptotic nuclear fragmentation for at least the 0 and 40 ug/mL MPC-4 treatments - as evidence?

3.2. Added section Results of flow cytometry experiments (Fig.2)

3.3. In the Intracellular signaling pathway regulation experiment section, we added a picture of protein expression (fig.4).

  • For Figure 3 – Can the author please include images/photographs of the blots as evidence?

(Same as above)

3.2. Added section Results of flow cytometry experiments (Fig.2)

3.3. In the Intracellular signaling pathway regulation experiment section, we added a picture of protein expression (Fig.4).

  1. RESULTS – Section 3.6
  • Can the author please include images/photographs of the tumor growth in the animal model as evidence?

No photos were obtained for this experiment. From the next experiment, we will add pictures.

  • For Table 3 – owing to the large standard errors/deviation, the differences perceived for tumor weight might not be statistically significant/meaningful. To resolve this, please perform statistical tests on the results and reinterpret the data. Also, please indicate the number of replicates done.

Added statistical analysis results.

  1. RESULTS – Section 3.7
  • Please indicate the number of replicates done.

 It is specified in section “2.7. Statistics and data processing”.

  1. DISCUSSION and CONCLUSIONS
  • Overall, this part seems poorly written, lacking coherence, and lacking discussion in relation to previous observations in the literature. Interpretation of results seems superficial. Please improve this.
  • More importantly, it largely looks like RESULTS – as most of the text simply repeated information already described in RESULTS. In some cases, the repeated statements were simply word-by-word repetition, which is not acceptable. Please improve this.
  • Paragraph 2 - “… in this study the efficacy of Momordica charantia linn fruits, … was experimentally investigated.” – This statement appears TWO times – in ABSTRACT and DISCUSSION – word-by-word. Please rephrase.
  • Paragraph 2 – “To that end, Momordica charantia linn supercritical extract (MPC-4) was … safety tests were performed.” – These two statements appear THREE times - in ABSTRACT, M&M, and DISCUSSION – word-by-word. Please amend.
  • Paragraphs beginning with the following: “First, as a result of the mouth-related bacteria inhibition and killing tests,…”, “Second, as a result of the cell proliferation inhibition experiment…”, “Third, as a result of the apoptosis-inducing effect experiment,…”, “Fourth, as a result of the intracellular signal transduction pathway regulation experiment,…”, “Fifth, as a result of the safety test, with regard to the cytotoxicity of the extracts in the cells,…”, and “Sixth, as a result of the animal model experiment, …” – these paragraphs are all redundant and repeated descriptions of information already provided in RESULTS. Please amend them.
  • Paragraph 10 - The statement “In this study, through the supercritical extraction method, the fact that … was presented.” - This statement appears TWO times – in ABSTRACT and DISCUSSION – word-by-word. Please rephrase.
  • Paragraph 11 – “Hot water Momordica charantia linn extract, pressurized Momordica charantia linn…, and intracellular signaling pathways regulating effects” – These are redundant and repeated descriptions of information already provided in RESULTS. No proper discussion/interpretation of the significance/meaning of results was presented.
  • Paragraph 13 – “expression of cleaved caspase 3 and Bak regulatory genes” – Please be careful to indicate that what was presented was protein expression results. Not gene expression results.

The "DISCUSSION and CONCLUSIONS" section has been completely rewritten with reference to the reviewer comments.

Reviewer 2 Report

This article investigated and compared the extracts of different plants prepared by different modes. The study is well designed. Before recommending this article for publication, there are some shortcomings for that should be resolve.

General comments

There many void and unclear sentences due to length of the sentences which must be clarify.

All the species names must be italic.

Properly revise the language of the MS.

Abstract

The authors elaborated abstract in a good way, but there are many sentences which are not clear. The sentences are very long such as “In this study, the efficacy of Momordica charantia linn fruits, Pistacia lentiscus extract, and Commiphora myrrha extracts, which are known to have large anticancer and anti-inflammatory effects, to prevent and treat oral solid cancer, in particular, inflammation or cancer in the oral mucosa was experimentally investigated author did not show specific results in the abstract.

The author should present some specific and quantitative results in the abstract.

Use italic font for species names.

In addition, the author should add future perspective of this study in the abstract section.   

Introduction

The introduction part is well written but some points must be discussing in detail. The author discussed about different ways of treatment of cancer and inflammation.

Provide the justification how plants and its extracts are better and easy to treat the said diseases.

The sentence is not clear “ In cancer patients receiving chemotherapy , stomatitis is one of the commonly appearing complications, which reduces the patient's immune function and ability to resist against bacteria infiltrating from the outside”

After the following sentence “Therefore, accurate early diagnosis, prevention, and treatment of stomatitis are necessary for cancer patients receiving chemotherapy”. Add 2-3 sentences on the uses and importance of medicinal plants against various diseases by citing relevant and recent literature.

https://doi.org/10.1016/j.jep.2021.114515, https://doi.org/10.1016/j.chnaes.2021.03.009, 

Materials and Methods

This section is well written.

The author can add figures/a plate of all figures of the studied species to convey clear picture to readers.

Results and discussion

Results are well written and presented. The discussion section lack comparison with relevant studies. The author is directed to compare the results with relevant studies provide proper reasons of the effects of these plants in various tests.   

Conclusion

Conclusion is well written but future perspective is missing. The author should add one to two sentences of future perspective for research point.

Author Response

Response to Reviewer 2 Comments

This article investigated and compared the extracts of different plants prepared by different modes. The study is well designed. Before recommending this article for publication, there are some shortcomings for that should be resolve.

General comments

There many void and unclear sentences due to length of the sentences which must be clarify.

All the species names must be italic.

Properly revise the language of the MS.

Abstract 

The authors elaborated abstract in a good way, but there are many sentences which are not clear. The sentences are very long such as “In this study, the efficacy of Momordica charantia linn fruits, Pistacia lentiscus extract, and Commiphora myrrha extracts, which are known to have large anticancer and anti-inflammatory effects, to prevent and treat oral solid cancer, in particular, inflammation or cancer in the oral mucosa was experimentally investigated author did not show specific results in the abstract.

The author should present some specific and quantitative results in the abstract.

Use italic font for species names.

In addition, the author should add future perspective of this study in the abstract section.   

All abstracts have been rewritten according to reviewer comments.

Introduction

The introduction part is well written but some points must be discussing in detail. The author discussed about different ways of treatment of cancer and inflammation.

Provide the justification how plants and its extracts are better and easy to treat the said diseases.

The sentence is not clear “ In cancer patients receiving chemotherapy , stomatitis is one of the commonly appearing complications, which reduces the patient's immune function and ability to resist against bacteria infiltrating from the outside”

It has been modified as follows.

(Stomatitis is one of the most common complications in cancer patients receiving chemotherapy. At this time, cancer patients lose their ability to resist externally infiltrating bacteria or normal bacteria, resulting in an inflammatory ulcer reaction in the oral mucosa.)

After the following sentence “Therefore, accurate early diagnosis, prevention, and treatment of stomatitis are necessary for cancer patients receiving chemotherapy”. Add 2-3 sentences on the uses and importance of medicinal plants against various diseases by citing relevant and recent literature.

The above sentence and the cited references have been added.

Zaman, W., Ye, J., Saqib, S., Liu, Y., Shan, Z., Hao, D., ... & Xiao, P. (2021). Predicting potential medicinal plants with phylogenetic topology: inspiration from the research of traditional Chinese medicine. Journal of Ethnopharmacology, 281, 114515.

https://doi.org/10.1016/j.jep.2021.114515, https://doi.org/10.1016/j.chnaes.2021.03.009, 

Materials and Methods

This section is well written.

The author can add figures/a plate of all figures of the studied species to convey clear picture to readers.

In order to present clearer results to the reader, the experimental items, contents, and experimental results have been modified as follows.

(Before revision)

2.2. Mouth-related bacteria inhibition and killing tests were deleted.

2.3. Cell proliferation inhibition test in oral cancer cells

2.4. Apoptosis inducing effect test

2.5. Intracellular signaling pathway regulation test

2.6. safety test

2.7. Tumor Xenograft Model

(After modification)

2.2. Cell proliferation inhibition test in oral cancer cells

2.3. flow cytometry experiment

2.4. Intracellular signaling pathway regulation test

2.5. safety test

2.6. Tumor Xenograft Model

2.7. Statistics and data processing

Results and discussion

Results are well written and presented. The discussion section lack comparison with relevant studies. The author is directed to compare the results with relevant studies provide proper reasons of the effects of these plants in various tests.   

In the “Results and discussion” section, we have supplemented the discussion points based on previous studies. This part has been rewritten to the core.

[29] Sasahira T, Kurihara M, Nishiguchi Y, Fujiwara R, Kirita T, Kuniyasu H: NEDD 4 binding protein 2-like 1 promotes cancer cell invasion in oral squamous cell carcinoma. Virchows Arch 469(2):163-172, 2016.

[30] Choi, S. E. (2019). Extraction method and physiological activity of high content oregonin derived from plant of Alnus sibirica Fisch. ex Turcz. Korean Journal of harmacognosy, 50(3), 165-174.

[31] Y.H. Kim, B.S. Park, The effect of eugenol on the induction of apoptosis in HSC-2 human oral squamous cell carcinoma, Journal of Korean Society of Dental Hygiene, vol. 15, no. 3, pp. 523-9, 2015.

[32] Pan G, Humke EW, Dixit VM: Activation of caspases triggered by cytochrome c in vitro. FEBS Lett 426(1):151-154, 1998.

[33] Hwang, E. M. Cell death induction effect of Yeoju extract on C6 glioma cells via reactive oxygen species and cell cycle regulation. 2018; Master's thesis, Hoseo University.

[34] Sanai, N., A. Alvarez-Buylla, and M.S. Berger. Neural stem cells and the origin of gliomas. N Engl J Med, 2005;353(8): 811-822.

Conclusion 

Conclusion is well written but future perspective is missing. The author should add one to two sentences of future perspective for research point.

The following has been added and supplemented to the Conclusion section.

These results suggest that the supercritical extract of Momordica charantia Linn., Pistacia lentiscus, and Commiphora myrrha can be applied as a preventive and therapeutic agent for inflammation and carcinoma of the oral mucosa in the future.

Round 2

Reviewer 1 Report

In the 4th paragraph of 2. Materials and Methods (page 3): “To experiment the efficacy of the extracts as such, mouth-related bacteria inhibition and killing tests,…” - Since the bacteria tests have been omitted in the revised manuscript, please omit this from this statement too.

“Inflammation” is mentioned in the title, abstract and discussion and conclusions. But I think none of the tests/assays performed directly/specifically measure the anti-inflammatory effect of the samples. Could the author please re-check and see whether it would be more appropriate to remove mention of “inflammation” especially in the title and conclusions?